# A Comprehensive Review of Electric Vehicle Charging Stations with Solar Photovoltaic System Considering Market, Technical Requirements, Network Implications, and Future Challenges

Ali Jawad Alrubaie [1], Mohamed Salem [1,*], Khalid Yahya [2], Mahmoud Mohamed [3,*] and Mohamad Kamarol [1]

1    School of Electrical and Electronic Engineering, Universiti Sains Malaysia (USM), Nibong Tebal 14300, Penang, Malaysia; ali.jawad@uomus.edu.iq (A.J.A.); eekamarol@usm.my (M.K.)
2    Department of Electrical and Electronics Engineering, Nisantasi University, 34467 Istanbul, Turkey; khalid.yahya@nisantasi.edu.tr
3    School of Engineering, Cardiff University, Cardiff CF24 3AA, UK
\*    Correspondence: salemm@usm.my (M.S.); mohamedmt@cardiff.ac.uk (M.M.)

**Abstract:** Electric cars (EVs) are getting more and more popular across the globe. While comparing traditional utility grid-based EV charging, photovoltaic (PV) powered EV charging may significantly lessen carbon footprints. However, there are not enough charging stations, which limits the global adoption of EVs. More public places are adding EV charging stations as EV use increases. However, using the current utility grid, which is powered by the fossil fuel basing generating system, to charge EVs has an impact on the distribution system and could not be ecologically beneficial. The current electric vehicle (EV) market, technical requirements including recent studies on various topologies of electric vehicle/photovoltaic systems, charging infrastructure as well as control strategies for Power management of electric vehicle/photovoltaic system., and grid implications including electric vehicle and Plug-in hybrid electric vehicles charging systems, are all examined in depth in this paper. The report gives overview of present EV situation as well as a thorough analysis of significant global EV charging and grid connectivity standards. Finally, the challenges and suggestions for future expansion of the infrastructure of EV charging, grid integration, are evaluated and summarized. It has been determined that PV-grid charging has the ability to create a profit. However, due to the limited capacity of the PV as well as the batteries, the Power system may not be cost effective. Furthermore, since PV is intermittent, it is probable that it will not be able to generate enough electricity to meet consumer demand.

**Keywords:** EV charging; photovoltaic systems; grid connectivity; standards

## 1. Introduction

The electrical power and transportation networks are beginning to integrate in a way that was before imaginable thanks to the EV's environmental, technical, and economic potential [1]. The main link between the two is the batteries, which power its EV's traction, control, lights, and air conditioning system. Charging the EV from the power grid, however, places additional load on the utility, especially during high demand hours. Prompting the charge of renewable energy sources is one method to mitigate the grid's negative impact [2]. The use of these clean energy sources is meant to reduce negative environmental consequences while also increasing the overall efficacy of the charging system [3].

Solar energy is becoming widely accepted as a competitive energy source of supplementing the grid due to the ongoing decline in photovoltaic (PV) module prices [4]. In addition, the PV system requires very low maintenance in terms of labor & fuel [5]. The development of energy converting technology, battery management systems, improved

installation methods, & design standards have all helped to significantly improve the application for PV to charge EVs (i.e., PVEV charge) [6].

A lot of the time, especially during the day, EV is left lazily sitting in the parking place, exposing to the full sun. It facilitates the expansion of charging options for EVs by making direct use of the "charging-while-parking" concept, it is meant to work in tandem with the standard "charging by halting" method. Installing a photovoltaic system on the parking garage's roof is one easy option for recharging these electric vehicles [7], while the owner of the vehicle is engaged in other activities [8]. The PV powered charging station offers a wide range of advantages, according to the authors in. The savings are particularly significant because charging takes place during the day, while load demand & electricity prices are their highest. Additionally, it has very low $CO_2$ emissions and small amount of fuel costs. This roof parking facilities are advantageous structural because they offer free shelter from sun & rain, which is important for countries with hot climatic conditions [9].

A PV-power, EV charge station uses PV generation as a secondary power point to recharge EVs, which will cut down on co-emission through fossil fuel-powered plants. In additional words, while the grid is down, EVs may still be charged using PV energy. In addition to reducing peak loads and improving microgrid stability via PV production and V2G, these technologies may also be used to reduce peak loads [10]. However, because to their mobility features, EVs are not the same as an energy storage system. Even if enough EVs are present at the charging station, an V2G may not occur, so would reduce peak power consumption or improve microgrid stability. Even though PV-powered EV charging stations have the potential to increase microgrid stability, there are a number of considerations that must be made [11,12].

The layout of a solar-powered EV charging station is shown in Figure 1. Solar panels, DC/DC converters, EVs, bidirectional EV chargers, as well as bidirectional inverters are the main components of a PV-powered EV charging station. Through a bidirectional inverter, the charging station is connected to the microgrid. The bidirectional inverter allows electricity from the grid to be delivered to the charging station [13,14]. Both bidirectional inverter as well as the microgrid have parallel connections to the local load. The solar array's output may either be utilised to directly serve local customers or added to the utility grid. As a means of monitoring the highest power point of a photovoltaic array, a direct current to direct current (DC/DC) converter is used (MPPT) [15]. The charging and discharging of electric cars are both within the jurisdiction of the bidirectional EV charger. To link EVs to a microgrid, a bidirectional inverter as well as reversible EV charging are required [16].

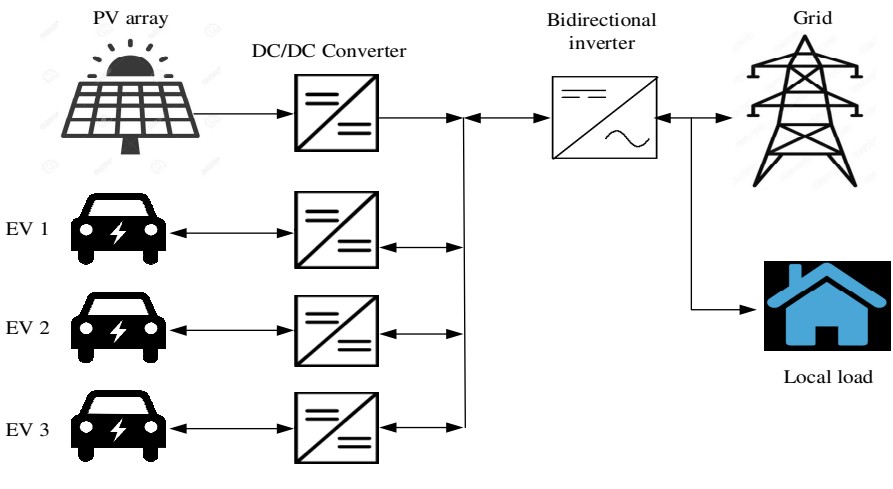

**Figure 1.** Structure of the investigated PV-powered EV charging station [17].

There are some review papers regarding EVs. Table 1, presents a brief review of some of the existing works along with the most important aspects that investigated in the articles under 7 domains including EV market review and analysis, technical requirements, EV charging infrastructure, grid concepts, overview of the current state of EV and analysis on important global standards, presenting future challenges and suggestions for the development of charging infrastructure, and EV-PV charging system. As it can be seen in the table, most of the mentioned papers are not considered EV-PV stations except Ref. [18] which it is not considered EV's market, and standards.

**Table 1.** A brief review on some of the related existing review papers carried out on electric vehicle.

| Domain | Ref [19]<br>Year 2022<br>Publisher Elsevier | Ref [20]<br>Year 2021<br>Publisher Elsevier | Ref [21]<br>Year 2022<br>Publisher Wiley | Ref [22]<br>Year 2021<br>Publisher IEEE | Ref [18]<br>Year 2022<br>Publisher IJRER |
|---|---|---|---|---|---|
| EV Market review and analysis | Technical characteristics of 17 EVs are compared. Also, a comprehensive analysis is carried done regarding the countries using EVs as well as introducing the most popular EVs. | The overall view of cost and characteristics of 10 EVs are compared. Also, the number of EVs in some countries are reported as well as the trend of banning of using internal combustion cars | The number of EVs used in some countries and their market's share are shown from 2015 to 2019 | There is no analysis regarding EV's market, while in a table a comparison about the technical characteristics of 4 EVs models is done | There is no analysis regarding EV's market |
| Technical requirements | Investigations of 3 charging techniques considering 5 aspects and the optimum site of charging stations are presented. Analysis of controlling infrastructures including the controlling architecture, centralized and decentralized controlling are presented. | Investigation of fast charging system (DC) considering converters and their instructions is done as well as a comprehensive analysis on different converters. | 13 articles are listed regarding the charge standards. Also, different aspects of controlling including harmonics and disharmonic, methods and architecture of charging controls. Discussion regarding the integrating of distribution sources with distribution network is the in the highlights of the study. | Improvement of off-board charger (EVs with IPT) and its infrastructure are investigated, since it addresses the need of a specific protocol for each EV. | Application of passive balancer for energy management of EVs considering various EVs and charge stations are investigated to analyse the output power of PVs. Also, considering: architecture of EV-PV, topology of converters, and 5 optimization methods regarding the improvement of the system |
| EV Charging infrastructure | Charging methods including BSS, CC, and WPT are discussed along with a comprehensive investigation of various conectors' protocols | On-off board charging systems are discussed along with a comprehensive analysis is carried done on converters as well as cost of charging stations in some countries | Charging infrastructures are discussed | Description of the topology of AC/DC and DC/DC converters. Power factor corrector, two stage onboard chargers and integrated onboard chargers are discussed. | Charge levels and their different moods along with vehicle coil detection system are presented |

**Table 1.** *Cont.*

| Domain | Ref [19]<br>Year 2022<br>Publisher Elsevier | Ref [20]<br>Year 2021<br>Publisher Elsevier | Ref [21]<br>Year 2022<br>Publisher Wiley | Ref [22]<br>Year 2021<br>Publisher IEEE | Ref [18]<br>Year 2022<br>Publisher IJRER |
|---|---|---|---|---|---|
| Grid concepts | Investigation of the positive and negative effects of integration of EVs with grid along with the role of distributors and aggregators on EVGI | An overall description of grid concepts along with the investigation of converter's topologies | Investigation of integration of distributed energy resources with grid considering DER standards and the role of data analysis on DER | A comprehensive description of the effect of charging stations on the grid considering RES, grid stability, demand-supply, assets, and current harmonics | Description of intelligent grid system along with V2G technology and intelligent transportation system |
| Overview of the current state of EV and analysis on important global standards | 25 different standards based on the standards' kind including connector, safety, charging teqniques are presented. Also, presenting the specific standard using by different manufactures | A comprehensive presentation of patents and projects of EV manufactures. Also, a comprehensive study is carried out on the charging stations standards | Investigation of 9 standards implemented in some countries | Charger standards are discussed | There is no description regarding standards |
| Presenting future challenges and suggestions for the development of charging infrastructure | Discussing the challenges regarding the integrating of grid, V2G technology, and range anxiety. Presenting the challenges and suggestions regarding the integration of grid. | Application of SiC and GaN in converters creates research rooms for fast charging and discharging | DR is considered as a challenge and the solutions are discussed. Challenges and barriers of EV adoption from different aspects including social, policy, and economy are discussed. | Introducing 7 cases that researchers focus on as challenges, such as V2G technology off board charging | Introducing of the different criteria on the charging time as a challenge |
| EV-PV charging system | There is no description regarding EV-PV | There is no description regarding EV-PV | There is no description regarding EV-PV | There is no description regarding EV-PV | A comprehensive study on EV-PV stations along with their architectures |

Considering more aspects of the research regarding EVs would cause to have a comprehensive source for readers. In this review paper, for the first time, all of the mentioned domains in Table 1 are systematically discussed to have an overview regarding other important related factors of EV-PV stations. The contributions of the present work are as follows:

- Electric Vehicle and Plug-in hybrid electric vehicles Charging Systems: global deployment of electric vehicle charging infrastructure, charging systems and their standardization, classification of electric vehicles charging levels, specifications, and standards, standards for electric vehicle charging as well as grid integration, electric vehicle charging standards, electric vehicle grid integration standards, safety standards for electric vehicle, electric vehicle integration in the power grid, modelling of grid-connected electric vehicle-photovoltaic system, electric vehicle smart charging using photovoltaic and grid

- Control Strategies for Power Management of Electric Vehicle/Photovoltaic System: intelligent energy management strategy, energy management strategy for smart home integrated with electric vehicle, and photovoltaic, control strategy for power electronic components
- Recent Studies on Various Topologies of Electric Vehicle/Photovoltaic Systems
- Challenges and Future Work Recommendations: modelling, optimization and control, issue on the integration with smart grid system, challenges and suggestions for electric vehicle charging

Figure 2 shows a flow diagram to see how the bibliometric review and research method and other parts of the current study is carried out.

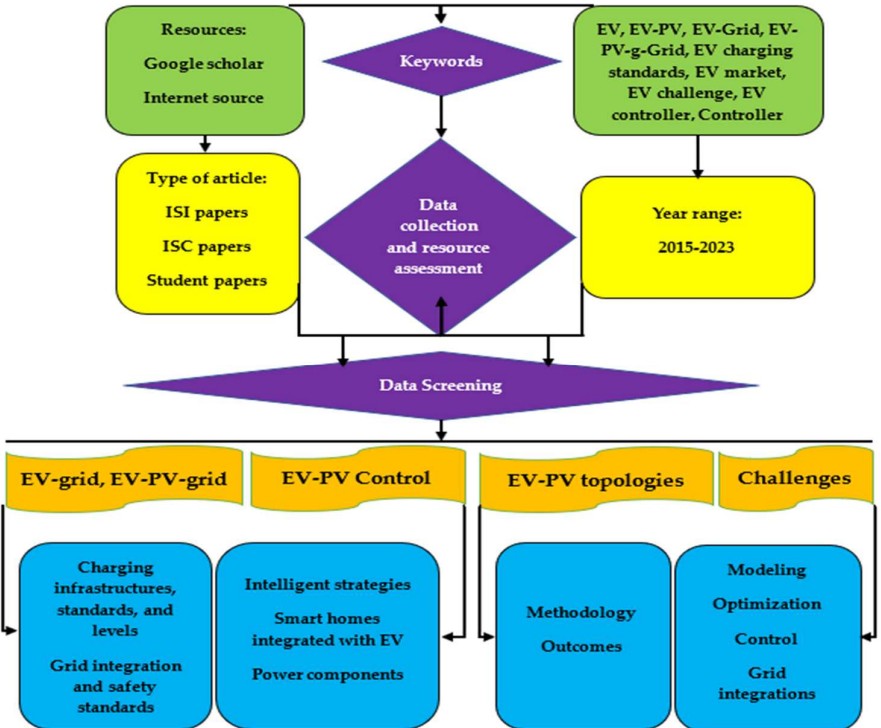

**Figure 2.** Detailed research method.

## 2. Electric Vehicle and Plug-in Hybrid Electric Vehicles Charging Systems

### 2.1. Global Deployment of Electric Vehicle Charging Infrastructure

To that end, the federal government of the United States has mandated that one million electric vehicles be in use by the year 2015, as well as numerous regulations have been put into place to encourage electrification of all spheres of public life. According to the Ontario Ministry of Transportation, using money of its Green Investment Fund, the Canadian state of Ontario plans to construct 500 EV charging stations (EVCSs) in around 250 places throughout the province by the end of 2017. It is estimated that by 2020, Germany will need around 70,000 public on-street charging spaces, according to the country's National Electric Mobility Platform (NPE) [23]. As a solution to the problems caused by China's current approaches to exploiting renewable energy and to keeping up with the ever-increasing energy needs of electric cars, the concept of placing a limited number to solar-powered charging stations to EVs is presented [24]. In May of 2017, the United States, Canada, France, Germany, Japan, the Netherlands, Norway, Sweden, and the United Kingdom formed the Electric Vehicles Initiative (EVI), a multi-govt policy forum with the goal of promoting the global adoption of EVs. South Africa, which joined the EVI in 2016 but remains a member, is actively involved in EVI operations, as do Korea and India [25]. The Indian government as well as major automakers have joined forces to promote e-vehicles as well as other clean fuel options in an effort to lower transportation-related pollution.

Thus, plan (NEMMP) 2020 for the nation's EV mission was initially suggested in 2013, and the following year it was signed into law. Faster Adoption and Manufacturing of (Hybrid &) Electric Vehicles (FAME) is another initiative in this space. The year 2015 also saw the announcement of India's intention to push the use of electric vehicles, and the India Scheme is introduced to support that effort [26]. It subsidises both utilities and their customers. The absence of charging outlets is now the biggest problem with India's infrastructure. There are still not a lot of places to charge your car around the country. Ninety-five percent of the world's electric vehicle stock and registrations are held by EVI members [27].

## 2.2. Charging Systems and Their Standardization

The classification of EVs charging levels, specifications, and standards, as well as in the next section, we will talk about controls strategies for EV/PV system power management. Finally, a review of the issues and the prospects for the future is provided, with an emphasis on the energy management system.

### Electric Vehicle Charging Standards

Full EVs enable total electrification of the transportation industry, in contrast, Electric Vehicle and Plug-in hybrid electric vehicles (PHEVs) are just partially electrified automobiles [28]. Various charging states are shown in Figure 3. In addition to high-power off-board chargers, Plug-in electric vehicles may also be charged continuously at any time of day or night using on-board level 1 or level 2 chargers (PEVs). An on-board integrated charge that can rapidly charge PEVs may combine the best features of conventional on-board and off-board chargers. Different charging levels had been defined by various organization throughout the globe, with an emphasis on the architecture of the vehicle and the kind and size of the battery utilized [29]. As a result, each car charges differently and behaves differently when charging. Therefore, the PV-EV charger must be designed in accordance with the aforementioned specifications.

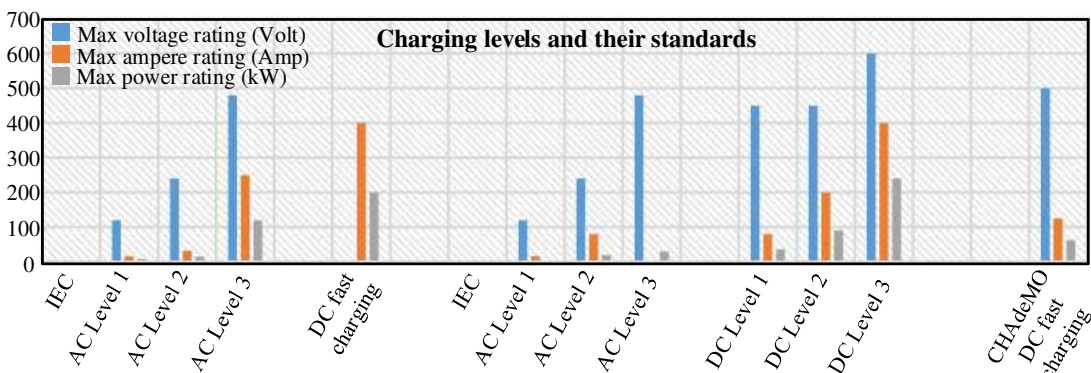

**Figure 3.** Graphical comparison of different charging levels and related standards [7].

## 2.3. Classification of Electric Vehicles Charging Levels, Specifications, and Standards

Conductive charging, wireless (or contactless) charging, and battery swapping are the three ways to refill an electric vehicle as classified in Figure 4. The most common and easiest way to charge anything nowadays is via conductivity [30,31]. In conductivity charging, the power source and battery are connected by a cable, whereas in wireless charging (WC), they are not. As opposed to conductive charging, WCs and battery shifting are currently under investigation and development [32,33]. The sections that follow will give further information on these technologies. Figure 4 provides a categorization of several charging methods.

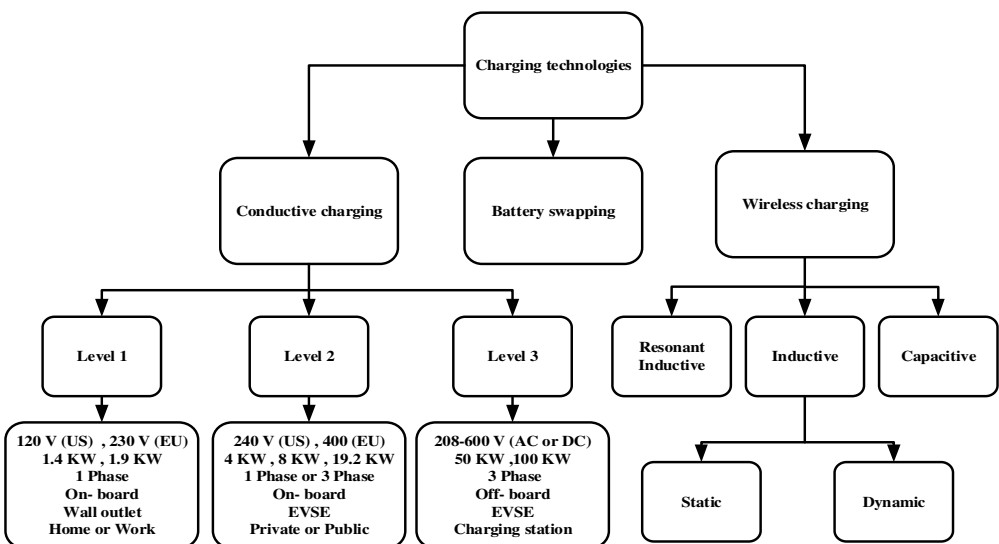

**Figure 4.** Classification of EVs charging technologies based on conductive charging, wireless charging, and battery swapping [34].

### 2.3.1. Conductive Charging

EV battery charger play crucial role for development of EVs since the adoption & societal acceptability of EVs is contingent on the ease of access of charging stations and public chargers. Various topologies of single phase & three phase EV charger are discussed [35]. A power factor adjustment unit, a DC-DC converter, and an AC-DC converter are the three components that make up this setup. There are two types of charger systems: on-board (inside the car, for slow charging) as well as off-board (outside the vehicle, for quick charging) (i.e., outside vehicle for fast charging) [36]. In addition, these chargers may be categorized as unidirectional or bidirectional. The gear for unidirectional charging is basic and merely permits electricity to flow from the grid to the electric vehicle. Bidirectional charging permits power to be transferred from the vehicle's charging station to the battery while driving on a public road; also known as "charging" to provide energy to a structure, the grid, or a home [37]. Potentially alleviating some of the stress experienced by EV owners and lowering the amount of energy storage required for onboarding may be achieved via the availability and development the EV charging infrastructure. The Society of Automotive Engineers (SAE) has established three different charging thresholds in its standard J1772 [38]. Level 1 and 2 chargers will be the norm for residential use, level 3 will be used by public charging stations, as reported by Electric Power Research Institute (EPRI) [38].

### Level 1 Charging

Due to lack of extra infrastructure and the ability to use any wall outlet, this method of charging is both the slowest and easiest. For Level 1, a typical 120 V/15 A wall outlet is utilized in the US. Only by on-board charger is offered. This EV takes a time to charge completely, although being less expensive than other charging levels. This charging level has the least negative effects on distributing networks because of its low power rating [39].

### Level 2 Charging

Level 2 charging utilizes 19.2 kW of charging power and 208 V or 240 V at currents up to 80 A. Because Level 2 requires less time to charge than Level 1, EV owners prefer it. For both private and public charging, specific electric vehicle supply equipment (EVSE) installation may be necessary. Electric vehicles like the Nissan Leaf have this much charging capacity built in [40].

Level 3 Charging

Level 3 is used for quick charging & functions like typical gas station (i.e., less than an hour of charging time) that may be put on major roads and highways. Due to the high charging power, which may surpass 100 kW it is only accessible as an off-board charger and is fed from a 3-phasing circuit with 480 V or greater voltages. It was obvious that level 3 charging is inappropriate for use at home. Its expensive installation raises the possibility of a problem [41]. Level 2 as well as Level 3 charging stations are expected to be widely used in public places including shopping centers, garages, restaurants, hotels, movie theatres, and other entertainment venues to facilitate rapid charging. Expensive charging power is advantageous to the perspective of charging time, but it may also result in peak demand, equipment overloaded on distributing network, and of high installation costs [42,43].

2.3.2. Wireless Charging

With WC, electric vehicles may be charged without a cord or any other direct physical contact here between power source and the battery. To reduce the cost, size, and environmental impact of EVs, advances in WC will reduce the amount of energy needed from the vehicles' onboard batteries. WC could replace conventional conductive charging in the future. The possibility exists to charge the batteries in electric busses using WC. It will function at various voltages (level 1, 2, & 3). 90% is at greatest efficiency for WC ever noted. WCs utilizes the inductive, resonant inductive, and capacitive technologies [34]. We'll go through inductive wireless charging (IWC) to give you an understanding of how the technology works. The AC electricity from the power grid is converted to DC power via the IWC's built-in AC/DC converter. It is then transformed once again into AC power and delivered to the transmitting (or main) coil at a high frequency. These parts are all located below under the roadway. The receiving (secondary) coil in the EV gets electricity from the transmitting coil via the air gap using electromagnetic induction [44–46]. After an AC/DC converter transforms the energy from an incoming power source into a usable direct current, the battery may be charged. Static inductive charging & dynamic inductive charging are two categories for WC. EVs must remain stationary while being charged using static inductive technology. Dynamic inductive charging, however, permits WC while an EV is in motion [47,48].

Existing WC implementations are geared at facilitating one-way flow of power from the grid to a vehicle, but developments in this area will enable electric vehicles to wirelessly withdraw the grid's energy to provide power. These benefits of this technology are user ease, cable-free operation, and electrical safety. The disadvantages of this technique are the poor power transfer efficiency between prices and the expensive infrastructure expenditures compared to conductive charging [49].

2.3.3. Battery Swapping

BSSs are charging station where empty batteries may be swapped out. A fully charged battery will soon replace the EV battery. Electric buses with large batteries that require a long time to charge using conventional conductive charging may employ battery swapping. The BSS or a third party that rents batteries to EV owners is required to maintain a sizable inventory for this technology [50]. The BSS is equipped with a distribution transformer, batteries, battery switching equipment, and AC/DC converters for charging the batteries. According to certain research, BSS might leverage bidirectional charges to provide electricity services using a V2G paradigm. Battery uniformity, significant infrastructure costs, and a huge BSS footprint are obstacles for this technology. A battery switching technology that can change the battery in 90 s was unveiled by Tesla Company in 2013 [51].

*2.4. Standards for Electric Vehicle Charging as Well as Grid Integration*

As a result of using EVs, the auto and power sectors have more room to expand. Every part of this new technology must be standardized if it is to be used consistently over the world. The standardization of EV charging may be divided into three categories according

on the Table 2: EV charging component standards, EVGI standards, & safety requirements. Component-level EV charging standards are developed by the International Organization for Standardization (ISO) and others, whereas ISO focuses on EV standardization as a whole [52,53].

**Table 2.** Classification table of comparison between different type of EVs charging levels, specifications, and standards.

| S.No | Charging Station | Voltage(V) | Power(kW) | Type of Vehicle (Wheels) | Type of Compatible Charger |
|------|------------------|-----------|-----------|--------------------------|----------------------------|
| 1 | Level 1 (AC) | 240 | ≤35 | 4, 3, 2 | Type 1, Bharat AC-001 |
| 2 | Level1 (DC) | ≥48 | ≤15 | 4, 3, 2 | Bharat DC-001 |
| 3 | Level 2 (AC) | 380–400 | ≤22 | 4, 3, 2 | Type 1, Type 2, GB/T, Bharat AD-001 |
| 4 | Level 3 (AC) | 200–1000 | 22 to 4.3 | 4 | Type 2 |
| 5 | Level 3 (DC) | 200–1000 | Up to 400 | 4 | Type 2, CHAdeMO, CCS1, CCS2 |

The requirements for EV discharging and charging into the grid have been specified. Electric vehicles serve as a DEV during grid charging as well as discharge (DER). As a result, DERs' grid connectivity regulations also apply to EVGI. The IEEE (Institute of Electrical and Electronics Engineers Engineers) as well as Underwriters Laboratories (UL) are two of the most influential organizations in the development of standards for grid connection [54]. The majority of the aforementioned organizations have established its safety requirements for EV charging & grid connectivity. On the other hand, the National Fire Protection Association (NFPA) and National Electrical Code (NEC) are heavily concerned with safety [55]. Subsequent sections elaborate on the norms and regulations established by such groups.

*2.5. Electric Vehicle Charging Standards*

EV charging infrastructure is included in a few international standards. Whereas IEC is commonly used in Europe, SAE & IEEE are employed by manufacturers headquartered in the United States [56]. The CHAdeMO EV charging protocol was developed in Japan. The Guobiao (GB/T) standard [52] is used in China for both alternating current (AC) and direct current (DC) charging, IEC standards [53] are identical to those of the GB/T for AC charging [55]. The Chinese National Committee for ISO as well as IEC developed this specification. Since the IEC [57] as well as SAE standards [38] for EV charging are the most widely used, we devote a lot of space to them here. From the specifications alone, it is evident that IEC61852 and SAE J1772 [38] are almost similar with the exception of certain language differences. While "level" is used to describe the intensity of an output in SAE, "mode" is the preferred term in the International Electrotechnical Commission (IEC) [57].

International Electrotechnical Commission standards

The IEC is a British organization that establishes norms for many forms of electrical, electronic, as well as related technologies.

- IEC61851 [52]. The IEC 61851 outlines the requirements for charging EVs and PHEVs using on-board and some off equipment utilising 1000 V AC and 1500 V DC supply voltages.
- IEC 61980 [52]. WPT systems up to 1000 V AC or 1500 V DC are covered under the IEC 61980 standard. This requirement applies to the WPT system that may be accessed using local storage facilities [57].
- IEC62196 [58]. IEC62196 specifies the plugs, socket outlets, vehicle connections, as well as vehicle inlets used for electrically conducting electric cars.

- SAE standards. The SAE is professional organization with headquarters in the United States that creates standards for engineering organizations in many sectors.
- SAEJ2293 [59]. For use with either an on-board or external power source, check out SAEJ2293 for detailed specifications. Part one of this standard, J2293-2, covers the information requirements and network architecture of EV charging, Part One of the J2293 standard discusses the energy needs and system design in three different operational scenarios (conductive AC charging, conductive DC charging, as well as inductive charging) [59].
- SAEJ1772 [38]. The current rating of circuit breakers and the voltage rating of chargers are both included in SAEJ1772. Both AC and DC are included in the definition of the standard, and each includes three levels. Almost all modern cars can receive Level 2 AC onboard charging at current that flows of less than 30 A. In terms of DC charging standards, the SAE DC level provides the fastest possible rate. However, a number of other elements, like as infrastructure and battery chemistry, have an impact on the actual charging rate [60].
- SAEJ1773 [38]. This standard outlines the minimal specifications for EV inductively linked charging systems. Inductive charging systems must be manually linked in accordance with SAEJ1773, which also specifies the criteria for the software interface [61].
- SAEJ2847 & SAEJ2836 [62]. These two standards, together with SAEJ1772, specify what electric vehicles and their charging stations must be able to communicate with one another. SAEJ2836 describes the scenario and provides the testing environment; SAEJ2847 details the necessary means of communication [62].
- SAEJ2931 [62]. This standard defines the requirements for digital interaction between electric vehicles, EVSEs, utilities, power utility interfaces, smart meter infrastructure, and home area networks. To make smart grid recharging of electric vehicles possible, a communication network must be constructed in accordance with SAEJ2931 standards.
- SAEJ2954 & SAEJ2954 [62] recommended practice (RP). When it comes to wireless charging, SAEJ2954 only supports level 2 (7.7 kW), while a recently published RP version claims to support level 3. (11 kW). For electric vehicle producers, the updated version will serve as a more uniform testing ground as well as infrastructure companies to evaluate the efficacy and validity of new products. This standard also incorporates autonomous charging, smooth EV parking, and payment setup [63].

### 2.6. Electric Vehicle Grid Integration Standards

IEEE1547, UL1741, & NFPA70 [32] are three accessible standards and codes. The following list of standards and codes highlights their key features.

- IEEE1547 [32] is a set of standards for integrating decentralized energy sources into centralized electrical grids. Specifically, performance, operation, testing, safety issues, and maintenance needs for interconnecting DERs are discussed, as well as DER installation on both secondary and primary network distribution networks with a PCC aggregate capacity of 10 MVA or less, it is appropriate to all DER technologies [64].
- UL standards [32] UL released a number of standards to address various DER grid interconnection issues. The criteria of power conversion equipment and related protective devices as they pertain to DER grid integration are covered in UL 1741 [32], the most relevant of these standards. Additionally, we adhere to UL62109, UL62109-1, UL62109-2 [32], as well as UL1741 SA [65].

### 2.7. Safety Standards for Electric Vehicle

This safety safeguard is a necessary component of grid connectivity and EV charging. Even though the majority of standardization bodies include safety requirements, NFPA and NEC place a strong emphasis on safety and security. Below is further information on the codes that these two organizations have created for EV charging and grid interconnection.

### 2.7.1. National Fire Protection Association Requirements

When it comes to educating the public about fire, electrical, and life safety, NFPA was an early adopter. The EV & its grid integration community has published its NFPA 70 [32] standard, which details wiring and safety procedures for electrical equipment on the customer side of a PCC [66].

They include:

- Electrical equipment and conductors mounted on or inside or outside of public or private buildings & other structure.
- Electrical wires that link the installations to an electrical supply as well as other external conductors & on-site machinery.
- Fiber optic cable.
- Structures utilized by the electric utility but not necessarily a component of power plant, a substation, or a control room [67].

### 2.7.2. National Electrical Code Standards

NEC [32] is different supplier of standards that focuses on EV safety measures. Additionally, it offers the specifications for EV charging hardware.

- NEC 625 [32], "Electric Vehicle Charging as well as Supply Equipment Systems", details the norms for EV charging infrastructure that is not part of the vehicle itself. Installation guidelines for EV charging station hardware are included. This includes things like conductors, connecting connections, including inductive charging devices, which are all part of the charging infrastructure connected either to feeder or branch circuits [68].
- NEC 626 [32]. The area of truck parking spaces is covered by this standard, which is labeled "Electrified Truck Parking Spaces." It specifies the requirements for the electrical apparatus and conductors used to charge trucks that are located outside of the vehicle. Circuit breakers, grounds, cable diameters, back feed prevention, & other requirements are among them [69].

### *2.8. Electric Vehicle Integration in the Power Grid*

Until recently, there was only tenuous connection between the transportation & electric power industries. The widespread electrification of transportation has significantly altered the established economic strategies of electric utilities [20]. Overall, EVs have given the electrical grid both considerable problems and advantages.

### Impacts of Electric Vehicle Integration on the Grid

EVGI has both positive and negative outcomes. The next subsections and Figure 5 illustrate the specifics.

- Negative Impacts: Electric utilities face a huge difficulty as a result of EVs. According to Table 3, excessive EV integration into the distribution network may have an effect on the distribution grid's stability, power losses, voltage and frequency imbalances, load profile, and component capacity [70].
- Positive impacts: It's true that having too many EVs upon that grid might cause issues with power quality degradation, an increase in peak demand, and challenges with power regulation, all of these concerns can be managed by adopting modern power management methods. The benefits of integrating EVs into grid in a strategic manner are summarised in Table 4 [71].

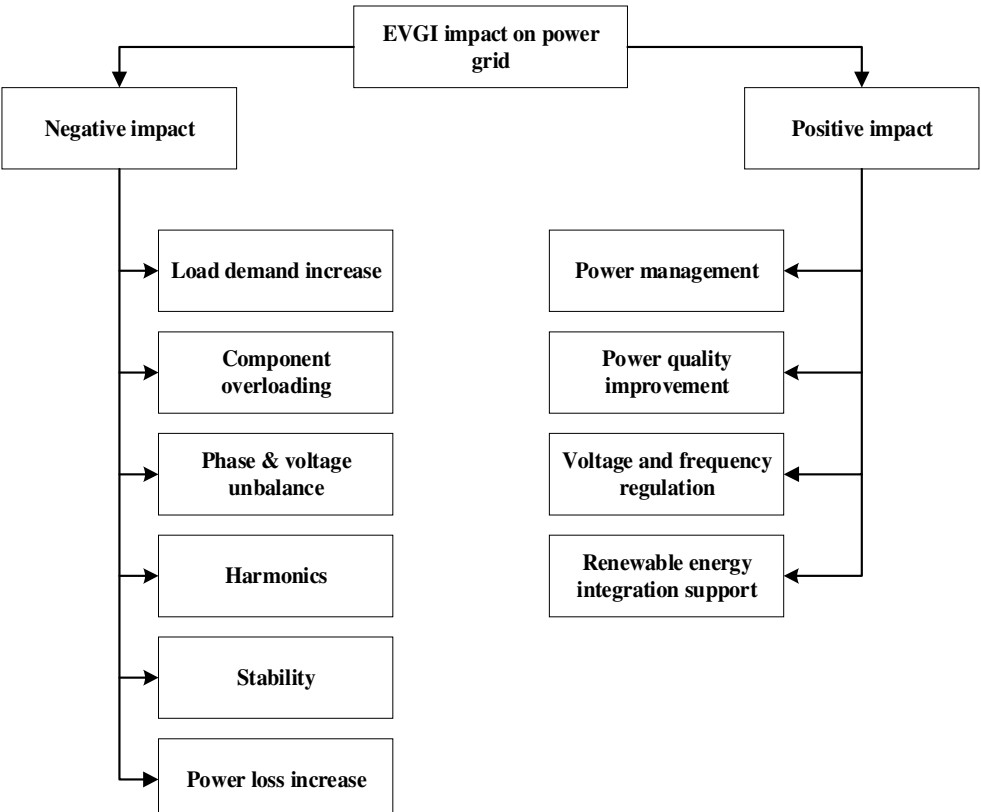

**Figure 5.** The positive and negative impacts of EV grid integration [32].

**Table 3.** Negative Impacts of EV grid integration along with the corresponding descriptions [32].

| Impacts | Description |
|---|---|
| Load demand increase | Up to 1000 TWh of extra loads (a 25% increase from present levels) may be added by EVGI. |
| | If electric vehicle charging is not controlled, the surge in demand during peak times might be a serious issue for utilities. |
| Component overloading | Extra-large EVGI values provide an incremental load demand that must be produced and communicated. The new demands are too much for the components of the current power system, which might lead to overloading and reduce transformer lifetime. |
| Phase and voltage unbalance | Since EV chargers are single phased, charging a lot of EVs at once might lead to phase imbalance. |
| Harmonics injection | Harmonics are created during the power conversion process by EV chargers since they are power electronic equipment as well as, if their penetration is larger, harmonize the grid. |
| | Although the THD level may increase as more chargers are being used, several studies suggest that it is less than 1% due to EV charging. |
| | A significant quantity of actual power is used as EVs become more and more integrated into the grid, which results in distribution system power loss. |
| Power loss | Given that 60% of the cars linked to the distribution system are electric vehicles, During off-peak times, power loss might increase by as much as 40%. |
| | Coordinated charging may increase the load factor of the electrical system while minimizing power losses. |
| | Power loss in the grid may be reduced by strategically placing charging stations. |
| | The power system becomes unstable when electric vehicle loads are added since they are nonlinear and consume lot of power rapidly. |
| Stability | More electric vehicles on the road makes the power infrastructure more vulnerable to outages and increases the time it takes to get things back to normal. |
| | EVGI has the potential to improve the stability of the electricity system if properly handled. |

**Table 4.** Positive Impacts of EV grid integration along with the corresponding descriptions [32].

| | |
|---|---|
| Power management | The use of planned charging and draining may improve power management. Discharging may be scheduled during peak hours to meet peak load demand. |
| Power quality improvement | Controlled EVGI may reduce voltage surges brought on by unchecked DER penetration. Flickering voltage may be reduced. When necessary, reactive power may be introduced. Uncontrolled DER harmonic injection may be minimized. Frequency control using grid frequency deviation regulation. |
| Regulation | Achieving voltage regulation via the production or consumption of reactive power. An equilibrium in power flow is achieved by the use of energy storage. Absorption of power that ramps up. An increase in the stability of isolated electric networks. |
| Renewable energy support | Using electric vehicles as energy storage might help smooth out the fluctuations in renewable energy production. Using electric vehicles as a buffer for renewable energy might reduce emissions and save money. |

*2.9. Modeling of Grid-Connected Electric Vehicle-Photovoltaic System*

2.9.1. Photovoltaic System

In this study, a diode serves as the *PV* module. As a result, a diode, a current source with a variable output, as well as a series as well as parallel resistance compensate the *PV* cell idea ($R_s$ and $R_p$). Evidence for the dependability of this paradigm already exists. To determine the output current ($I_{pv}$) ones from a *PV* system, apply the following [72,73]

$$I_{PV} = I_{Ph} - I_{sat}\left(e^{q(V_{PV} + I_{PV}R_s)/(NkT_{PV})}\right) - (V_{PV} + I_{PV}R_s)/R_P \tag{1}$$

$$I_{sat} = K_1 T^3 e^{\left(-\frac{qV_g}{kT}\right)} \tag{2}$$

where *K* represents the Boltzmann constant, $I_{sat}$ represents the saturation current of the diode, Ko and K1 represent constants that are dependent on the features of *PV* cells, in the *PV* model, N represents the input impedance of the an diode, *q* represents an elementary of an electron, and *T* represents the temperature at which the *PV* cells are active [74].

A DC/DC converter completes the virtual photovoltaic system. This part controls the PV system's output voltage as well as makes the *PV* voltage compatible with MVDC bus voltage so that the *PV* system can run in maximum power point tracking mode. The DC/DC converter might be switching to a different mode (DC bus voltage sustaining mode) in order to maintain a constant MVDC bus voltage, resulting in a reduction in the quantity of power provided by the *PV* modules [75].

2.9.2. Electric Vehicle as Well as Energy Storage System Batteries

The SimPowerSystems toolbox of Simulink's model has been used to simulate these batteries for both EVs & energy storage system (ESS). One resistance and one variable voltage source are coupled in series to make up this type. As a result, battery voltage may be determined by the steps below [8].

$$V_{bat} = E_{bat} - I_{bat}R_{int} \tag{3}$$

where $R_{int}$ is the internal resistance of the battery, $I_{bat}$ is the current through the battery, and $E_{bat}$ is the open-circuit voltage, that changes with the state of charge or discharge. To prevent the battery from being damaged by excessive charging or discharging, one of the most crucial battery metrics to be monitored is the state-of-charge (*SOC*) [76]. Inferring it from following expression:

$$SOC\ (\%) = SOC\ (\%) - 100\left(\frac{\int I_{bat}dt}{Q}\right) \tag{4}$$

$Q$ = maximum battery capacity.

### 2.9.3. Direct Current/Direct Current Converters

All of the DC/DC converters' dynamic behavior has been modeled using average-value equivalents. These models are often used to investigate lengthy simulation situations because it may be possible to recreate the converters' dynamic behavior faithfully. This DC/DC converter type consists of a voltage source as well as a current source that may each be adjusted independently [14]. Depending on the converter being utilized, the duty cycle links the input, output voltages & currents. For a complete rundown of a DC/DC converters found in the tested systems, see Table 5.

**Table 5.** Summary of the dc/dc converters found in the tested systems and functions [77].

| Component | Converter | Energy Flow (From–To) | Transfer Function |
|---|---|---|---|
| PV | Unidirectional-boost | (PV–DC bus)–boost | $M_{DC} = \frac{V_o}{V_i} = \frac{V_i}{V_o} = \frac{1}{1-D_C}$ |
| EV | Unidirectional-buck | (DC bus–EV)–buck | $M_{DC} = \frac{V_o}{V_i} = \frac{V_i}{V_o} = \frac{D_C}{1-D_C}$ |
| EV | Bidirectional | (ESS–DC bus)–boost | $M_{DC} = \frac{V_o}{V_i} = \frac{V_i}{V_o} = \frac{1}{1-D_C}$ |

### 2.9.4. Grid Connection

The link to the neighboring 20 kV AC grid is made possible via a delta-wye transformer as well as a three-phase IGBT inverter. This components' representation based on models created. A Space-Vector PWM modulator is used to drive the inverter under control [78].

### 2.10. Electric Vehicle Smart Charging Using Photovoltaic and Grid

Numerous research has looked at the benefits of PV-based EV charger systems. Reference highlights the benefit of charging the EV with PV and explains how it enables for higher penetration of both PV as well as EV. Additionally, EVs may lessen the consequences of excessive PV production. Reference provides a case study of Columbus, Ohio, to demonstrate that solar-powered EV charging is both cheaper and less polluting than traditional grid-based EV charging [79]. The grid integrated PV system outperforms the other two systems economically, a case study contrasting three different approaches to recharging electric vehicles (grid alone, PV just plus battery storage, as well as grid integrated PV). The authors of reference discuss the use of PV power and EVs as a battery system to reduce peak grid demands. Studies like this show how much better PV-based EV charging is compared with grid-based EV charging. Numerous papers discuss alternative charging algorithms and what role they may play in meeting the financial, technological, and societal goals of PV-based EV charging [80].

The problem formulation determines the kind of optimization model. Convex type problems (linear, mixed-integer, and quadratic) often have cheap processing costs and may reach optimum solutions [81]. Optimization strategies of the meta-heuristic kind, such the Genetic Algorithm and the Particle Swarm Optimization, may be used to solve non-convex problems and provide solutions that are close to optimum while requiring little computing effort. With limited data and processing resource needs, the rule-based algorithm or heuristic type optimization techniques may give adequate solutions for arbitrary instantaneous occurrences (such as the plugging in/unplugging of EVs or the change in PV power, for example) [82].

The reason why PV-based EV charging systems for homes and offices have received more attention than those used in commercial settings is that they need less complex analysis and flexible integration inside the distribution system. Additionally, the majority of research on smart charging focuses on particular ways to improve EV grid integration, for example, variable rates of pricing, market participation, and supplementary offerings A complex system with many different components is needed to simulate the real-life implementation [83].

## 3. Control Strategies for Power Management of Electric Vehicle/Photovoltaic System

The power flow between the system's four key components must be investigated in order to create the suggested system layout. Electrical grid, PV sources, battery storage, as well as EV charging load are the major components. Power flow management can be used to establish the demand for and size of a bidirectional power flow energy automated device. Consequently, the study introduces their applicability on the examined application in an effort to answer the power flow management issue [83,84].

Heuristic algorithms that take into consideration loaded demand, PV insolation levels, including off utility hours are often used to estimate power flow in energy PV/battery systems. However, the proposed system's solution is made considerably more difficult by a dynamic grid pricing [85,86]. Using the streamlined heuristic concepts to operate the PV/battery system will provide operational cost solutions that differ considerably from of the least cost operation underneath a dynamic grid pricing model. As a result, the pace of research in this field has quickened [87].

The intended architecture for power flow management must allow for non-linear operations. This enables the designed topology to be used to various operational circumstances. To enable the system to function properly under mismatched circumstances and predicting deviations, the topology must feature an online error compensation phase [88,89]. Last but not least, the online optimization step should be built with a short calculation time so that it may be quickly incorporated into real-time controllers. To get over these issues and carry out the suggested optimization, research on smart methods of energy management and flexible power distribution is urgently required [90].

### 3.1. Intelligent Energy Management Strategy

In its present form, the EV-PV charger should be able to charge an electric vehicle using solar power, but it has no intelligence of its own [91]. According to forecasts, the cost of electricity will be lowest in the morning hours, this being an ideal moment to plug in your electric vehicle to the grid for a recharge. While solar charging is most effective during sunny afternoons. Smart charging algorithms are required for the control of EV-PV systems to be realized [92].

Every car has a predictable period of accessibility as a load, and this condition of charging the automobiles at parking lots has been taken into consideration. While parking the car, the user specifies the billing interval and makes any other relevant price/billing type/etc. selections.

Capacity, chemical composition, open-circuit voltage, battery state of charge, ambient temperature, and so on are all characteristics of a vehicle. Every car will allocate the power to the system based on the user's preferences and the battery's characteristics. So, in order to sum up, dynamic power management of loads takes into account user preferences and load variables to optimize energy consumption [93,94]. Improving demand-side power management, control mechanisms, as well as consumer preferences for utility services are the cornerstones of grid design. A specific system was designed to show how the smart grid works [95,96]. The skeleton of the whole system is shown in Figure 6. All of the advantages outlined and taken into account during the system's design are realized via battery loads, charging, and connection to the power grid all fall within the purview of energy management. Every parking deck is controlled by the Intelligent Energy Management Strategy (IEMS), and each parking deck is made up of different loads [97,98].

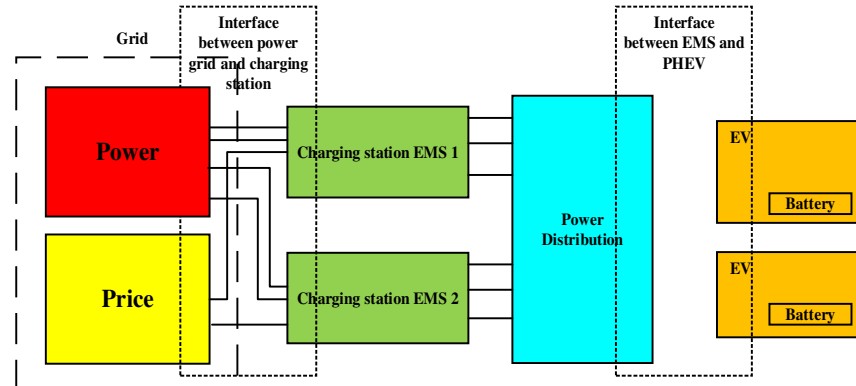

**Figure 6.** Architecture of the intelligent energy management system considering the performance of a smart grid [99].

### 3.2. Energy Management Strategy for Smart Home Integrated with Electric Vehicle, and Photovoltaic

To limit load power curve volatility and associated energy costs in homes with PV and EV, energy management measures may be essential. With the proposed method in [99]., electricity may be transferred from the grid to vehicles (G2V), photovoltaics to vehicles (PV2V), as well as vehicles to houses (V2H), therefore reducing daily energy expenses and smoothing out graph of smart home electrical consumption [100]. When PV power generation is unavailable (PV power is equal to 0), stage A of the proposed control technique has three modes of operation, while Stage B has five modes of operation (PV power greater than zero) [101]. The primary objectives of the approach are to manage the energy drawn from the grid and/or PV and the energy given to a load demand by EVs, and to regulate the charging and discharging cycles of EVs. Maintaining the load curve inside the optimal energy consumption range will reduce power bills, the following goals may be attained [102].

- Find out when the price of power is at its lowest and highest.
- Determine when the residence under study uses the most and the least power.
- To save energy costs and increase valley capacity, at moments of low power demand as well as cheap energy costs, charge electric cars from the grid utility.
- Manage the battery's level of charge to avoid overcharging and overdischarging it when charging and discharging, respectively [103].
- Learn when PV production is inversely proportional to the amount of needed power that can be supplied by photovoltaics, and use those periods to charge your electric vehicle from the extra energy you've generated.
- Electric car discharge during peak demand periods has the potential to minimise peak demand [104].

The following data is used as inputs in the proposed method to accomplish the aforementioned aims: the time the EV Tin arrives, the time the EV Tout leaves, SOC of an electric vehicle battery at time t, maximum (SOCmax) and minimum (SOCmin) levels of charge, average power load (Pavg) and average price signal (Cavg), and time-of-use price signal (C) are all variables to consider; the profile of solar power production Ppv (t); and so on [105].

### 3.3. Intelligent Energy Management Strategy

In its current version, the EV-PV charger can take in solar energy and charge the EV, but it does not have any specialised knowledge on how to do it. The cost of electricity is predicted to remain low throughout the morning; hence, charging an EV from the grid is more profitable in the morning. While bright afternoons are advantageous for solar charging. A smart charging algorithm is required for the control of an EV-PV system [106,107].

As each parked car is available for a certain amount of time as a load, the pricing situation in carparks has been investigated. The user chooses the charging period, price, and kind of charge at the time of parking [108,109].

A vehicle's capacity, chemistry, open-circuit voltage, charging levels, the state of charge, temperature, and similar factors are all additional features. Every car will allocate power to the system based on user selection and battery parameters. To summarise, the system's energy usage is maximised by taking into load factors and customer preferences into consideration while implementing dynamic power management [110]. Control methods, advanced demand-side power management, as well as customer service preferences are some of the primary drivers behind grid layout. A specialised system was designed to demonstrate the benefits of the smart grid. All the advantages outlined and results expected by integrating a power grid, charging infrastructure, and battery loads into an existing energy management system. Each parking structure under the IEMS's jurisdiction holds a broad range of cars and must adhere to its rules [111,112]. The IEMS is automatically updated with information about available power and cost from the grid's perspective at a given time interval. This approach assists IEMS in monitoring several converters at the same time in order to make real-time choices [113–116].

### 3.4. Control Strategy for Power Electronic Components

Power converters are used to link all the parts of the system to a DC power source. The maximum output of a PV array is constantly monitored by the boost converter. When it comes to performance as well as battery life, a bidirectional DC/DC conversion is used for charging and charging the battery. Electric cars need two separate control systems in order to work with a broad variety of battery sizes. We'll look at two types of controllers: (1) the I-V controller, as well as (2) the average sliding mode control [117].

### 3.4.1. I–V Controller for Electric Vehicle

The EV runs in a current loop mode to enhance an EV's charging capacity. Charging the EV requires just a correct current reference control algorithm [118]. The system's health determines how often this resource is refreshed. Feedback from a comparison of a current drawn from a PI controller is able to maintain a steady-state error of zero by connecting the batteries of a hybrid or electric car to a current reference illustrated in Figure 7 [119,120].

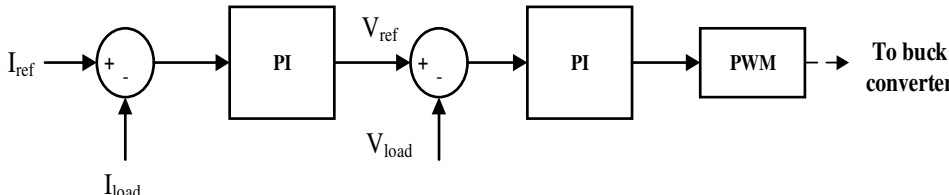

**Figure 7.** Block diagram of I-V based controller for EV from input to output [100].

Meanwhile, the EV's battery voltage is compared to a value computed by the PI controller. The DC/DC converter is controlled by pulses generated from an external PI block's output. The electric vehicle may be charged to a maximum of 6 kW when overall power generation exceeds consumption [121].

### 3.4.2. Average Sliding Mode Controller to Electric Vehicle

Nonlinear control is used because conventional linear controllers like proportional-integral-derivative (PI) as well as proportional-integral (PID) are inadequate when dealing with substantial input as well as load voltage fluctuations [122]. Despite variations in the system's features and load, sliding mode management maintains stability and resilience. To improve performance, multi-loop control is used instead of basic loop control, however designing the controller is difficult, especially for higher order converter topologies [123]. Sliding mode control and hysteresis control are two examples of nonlinear control ap-

proaches that provide very accurate models. To regulate DC/DC boost converters, several current controllers are employed. In order to fine-tune the current characteristics and attain the targeted dynamic performance, average sliding mode control (SMC) is used, as shown in Figure 8. For this reason, the average SMC reference current is derived from PI current control block's output [19].

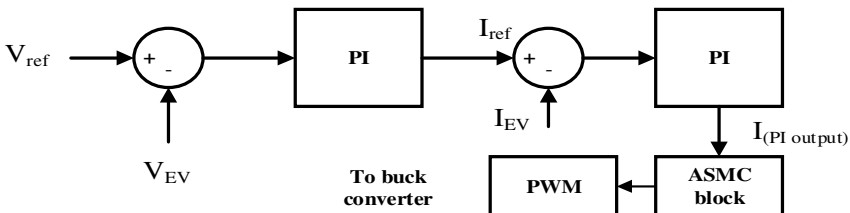

**Figure 8.** Block diagram of average SMC from input to output [100].

The varying switching frequencies make it very sensitive to overall SMC noise. Few regulated options exist to address the issue of variable switching frequency. After the median SMC has established the allowable signal range, its variable switching frequency is locked in at a fixed value. Figure 9 depicts a typical SMC building block [124]. Output current as well as voltage from the EV are calculated and compared to standards before being sent to the PI controllers [125].

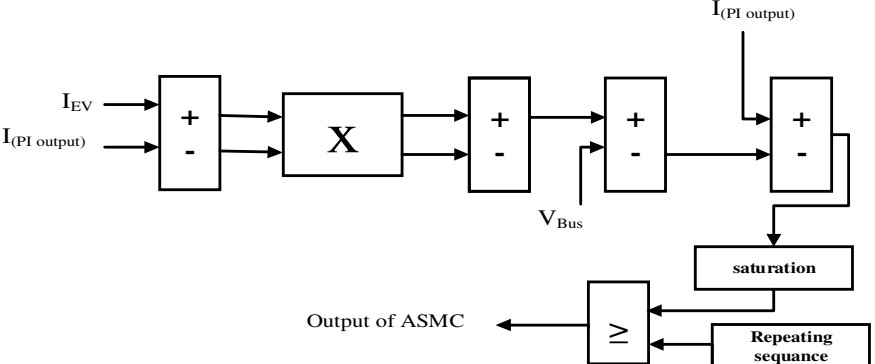

**Figure 9.** Average SMC for current control in its typical building shape [100].

## 4. Recent Studies on Various Topologies of Electric Vehicle/Photovoltaic Systems

Table 6 discuses the recent studies of the EV-PV systems.

**Table 6.** Summary of Topologies of EV/PV Systems based on the recent studies and their methodologies and outcomes.

| Reference | Methodology | Outcomes |
|---|---|---|
| [126] | a high gain, fast charging DC–DC converter and a control algorithm for grid integrated Solar PV (SPV) based EVCS with battery backup | The proposed converter and its control algorithm's performance are investigated in three different modes using MATLAB/Simulink tool and the simulated results are validated with Real-Time Digital Simulation (RTDS) in OPAL-RT. The observed results meet the Power Quality limits of an IEC61000-3-2 [52] standard and met the level-4 dc charging standard IEC61851 [32] |

**Table 6.** *Cont.*

| Reference | Methodology | Outcomes |
|---|---|---|
| [127] | minimizing the cost incurred due to energy losses in an IEEE-37 bus system integrated with a commercial EVCS located in Qatar. The Particle Swarm Optimization (PSO) algorithm is used for the efficient location allocation of EVCS. | The system is analytically examined through the Thukaram Load Flow Algorithm and investigations are conducted to observe the beneficial impacts of load balancing between RES and utility |
| [128] | conceptualizes the interconnection of these components through a 750 V DC nanogrid as against a conventional three-phase 400 V AC system. The factors influencing the performance of a DC-based nanogrid are identified and a comparative analysis with respect to a conventional AC nanogrid is presented in terms of efficiency, stability, and protection. | It is proved how the minimization of grid energy exchange through power management is a vital system design choice. Secondly, the trade-off between stability, protection, and cost for sizing of the DC buffer capacitors is explored. The transient system response to different fault conditions for both AC and DC nanogrid is investigated. |
| [129] | , an application mode of EV charging network and distributed photovoltaic power generation local consumption is studied. The management idea of two-layer and four model has been established, including the regional distributed photovoltaic output model, electricity consumption model, EV consumption model, and regional grid load dispatching model, which can realize the scheduling of the energy flow formed by photovoltaic, induce the charging of EVs, and make the photovoltaic consumption in office building areas and residential building areas complementary | This mode is intended to guide the consumption of new energy through economic leverage, which can realize the unified regulation of distributed energy convergence, consumption and storage. |
| [130] | mainly focuses on the various standards for EV, PV systems and their interconnection with grid-connected systems. | For a better operation of EV for domestic and commercial use, it is necessary to follow certain international standards that are published and proved to be efficient. Some of the important standards to be followed in the design and manufacturing of EVs. |
| [131] | presents the control of a single phase EVCS powered by a wind energy generation system (WEGS) and a PV array that are installed in houses for EV charging and household supply. | Test results have proven the satisfactory operation of the CS for supplying the domestic loads while accomplishing the primary task of charging the EV. |
| [132] | r reviews the state-of-the-art literature on power electronics converter systems, which interface with the utility grid, PV systems, and EVs. Comparisons are made in terms of their topologies, isolation, power and voltage ranges, efficiency, and bi-directional power capability for V2G operation. | A brief description of EV charger types, their power levels, and standards is provided. It is anticipated that the studies and comparisons in this paper would be advantageous as an all-in-one source of information for researchers seeking information related to EV charging infrastructures. |

## 5. Discussion and Conclusions

This paper provides an overview of PV-EV charging system technology, operation, and status. In addition, it provides information on the principles of electric cars, batteries, and a description of PV. To prove the technological and economic feasibility of PV-grid as well as PV solo charging, a case study is carried out by contrasting them with grid-only charging. It has been determined that PV-grid charging has the ability to create a profit. However, due to the limited capacity of the PV as well as the batteries, the Power system may not be cost effective. Furthermore, since PV is intermittent, it is probable that it will not be able to generate enough electricity to meet consumer demand.

## 6. Challenges and Future Work Recommendations

### 6.1. Modeling, Optimization and Control

In spite of the various efforts that have been done on PV–grid charge techniques, it is essential to the alternating nature of PV be recognized. Because of the unpredictability of the sun irradiation, there is no way to know how long the charcoal will keep its consistency. As a result, it is of the utmost importance to design the system to have an energy management function that is optimum. This development of reliable forecast model for power production of PV systems is one topic that is of interest [133]. This may be linked with an accurate model of the cost of grid energy (which is basing on dynamic tariff structures), which will guarantee that the owner of the charging station obtains the highest possible return on their investment [134]. The following works are evidence of the growing interest in the various energy management systems.

Static charging requires electric car owners to notify the station owner in advance of the anticipated demand for charging their vehicles. In more realistic dynamic charging situation, an electric vehicle (EV) may enter and depart the station at any moment, without the owner of the station having previous knowledge of this [135]. In order to accomplish this goal, optimum solutions basing linear programming & heuristic algorithms used, one for static issues & the other for the dynamic ones, respectively [136]. Despite the fact that the dynamic scenario presents a more accurate representation of reality, the answers provided by the static issues may serve as a standard against which performance can be measured. The optimization of electric vehicle charging has a number of possibilities for the use of approaches from soft computing [137].

The optimization of the photovoltaic charging system utilizing the soft computing approaches has a number of difficulties, despite the fact that these methods are successful in themselves. For instance, the fuzzy logic controller is simply appropriate for use in a system that has a limited number of charging outlets and electric vehicles that need to be charged [138]. The rule table will get more complex as well as the algorithm will take longer time to execute if there are a large number of energy sources and EVs. In addition, achieving exact tuning of the fuzzy variable quantity is quite challenging, yet with the assistance of methods such as the genetic algorithm [139].

### 6.2. Issue on the Integration with Smart Grid System

It has been determined that smart grid technology, which has renewable energy sources & electric vehicles play a vital part, will be that dominant trend in the fields in future power systems. For instance, electricity that is produced by PV may alleviate some of the strain placed on the grid, especially during peak hours [140]. However, utilities' worries about potential unintended consequences, such as impacts on system reliability, power quality, protection, as well as grid synchronisation, have increased in tandem with the widespread use of renewable energy sources. This is all tied up with PV's status as an intermittent energy source. It is regarded to be a big problem for the smart grid system to handle these qualms in PV–EV charging, which gives intriguing opportunities for additional exploration [139].

### 6.3. Challenges and Suggestions for Electric Vehicle Charging

- Range, refill time, and cost all play a role in determining how popular electric cars are. The availability of charging stations for EVs is crucial to these considerations. Below, we highlight some of the difficulties and recommendations associated with EV charging [141].
- There are not universal manufacturing requirements for charging devices. For instance, there are different standards for charging connectors in Japan, the US, and Europe. Homogeneity in charging standards and equipment may save costs and increase market acceptance of EVs.

- Currently, not each EV models can handle every charging levels, & Warning: not all public charging stations can handle high-power devices. Because of this, those who own electric vehicles have a hard time finding enough charging infrastructure [142].
- Currently, fast charging station users must pay a set monthly demand fee, which discourages EV owners from utilizing their vehicles since they are unable to charge them on demand based on a variable power tariff. Modifying the fixed demand fee policy may be able to alleviate EV owners' complaints [143].
- The charging facility layouts of the charging stations vary since they remain installed by various businesses in various locations. The users find it difficult to adjust to different charging facility layouts. The popularity of EVs will rise with a uniform charging facility structure comparable to ICEV refuelling stations [144].
- Private fast-charging facilities, such those in homes, are still difficult to set up and often need permission from local service companies & govt. This drawn-out procedure discourages EV owners from building their own private fast charging infrastructure for satisfying their needs.
- It's important to strategically deploy EV charging stations along major thoroughfares and in urban areas. Owners of EVs are concerned about the lack of planning for charging stations outside of major cities [145].
- Renewable energy sources, including solar or wind energy, may be used in charging stations. Such charging stations demand a lot of room and expensive design and execution. Vacant lots near roadways are excellent locations for renewable energy-powered EV charging stations [146].

**Author Contributions:** Conceptualization, A.J.A., M.S. and M.K.; methodology, K.Y. and M.M.; writing—original draft preparation, A.J., M.S. and M.K.; investigation, M.M. and M.K.; resources, A.J.A. and M.S.; writing—review and editing, M.S., M.M. and K.Y.; visualization, M.K. and A.J.A.; supervision, M.S. and M.K.; funding acquisition, M.M. All authors have read and agreed to the published version of the manuscript.

**Funding:** This research received no external funding.

**Institutional Review Board Statement:** Not applicable.

**Informed Consent Statement:** Not applicable.

**Data Availability Statement:** Data sharing not applicable.

**Conflicts of Interest:** The authors declare no conflict of interest.

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
