# Peer review of "A Comprehensive Review of Electric Vehicle Charging Stations with Solar Photovoltaic System Considering Market, Technical Requirements, Network Implications, and Future Challenges"

_sustainability, doi:10.3390/su15108122_

Round 1

Reviewer 1 Report

1.      The title of the manuscript does not exactly reflect its content.

You examine the present electric vehicle market, technical requirements, charging infrastructure, and grid implications. You also give an overview of the present EV situation and an analysis of significant global electric vehicle charging and grid connectivity standards. Lastly, you present brief challenges and suggestions for future expansion of the infrastructure of electric vehicle charging, and grid integration.

This means there is no specific focus on solar photovoltaic systems. Please consider modification of the title

2.      The implication of this research is not clear in the abstract and introduction sections. Please provide more information and be more specific on the novelty of the work.

3.      There are many recent similar review papers in this domain. What is the difference between your review and the following papers? If there is a significant difference you should put it in the introduction section. Otherwise, there is no point in publishing this paper.

·         Mastoi, Muhammad Shahid, et al. "An in-depth analysis of electric vehicle charging station infrastructure, policy implications, and future trends." Energy Reports 8 (2022): 11504-11529.

·         Rajendran, Gowthamraj, et al. "A comprehensive review on system architecture and international standards for electric vehicle charging stations." Journal of Energy Storage 42 (2021): 103099.

·         Sachan, Sulabh, et al. "A comprehensive review of standards and best practices for utility grid integration with electric vehicle charging stations." Wiley Interdisciplinary Reviews: Energy and Environment 11.3 (2022): e424.

·         Khalid, Mohd Rizwan, et al. "A comprehensive review on structural topologies, power levels, energy storage systems, and standards for electric vehicle charging stations and their impacts on grid." IEEE Access 9 (2021): 128069-128094.

·         Kannan, Geetha Palani, and S. Usha. "Critical Review on and Analysis of Solar Powered Electric Vehicle Charging Station." International Journal of Renewable Energy Research (IJRER) 12.1 (2022): 581-600.

4.      The method of literature search and inclusion criteria should be included in the methodology section. This should include year range of publications and online databases used (such as the Web of Science, Scopus, ACM Digital Library, Emerald, IEEE Xplore, SAGE Journals, ScienceDirect, SpringerLink, Taylor & Francis Online, MDPI, etc). This will enable the reader to replicate the data when the need arises.

5.      There are many places where you defined an abbreviation more than one time. E.g., in line 252 you defined “International Electrotechnical Commission (IEC)” and you defined it again in lines 254 and 256. Please review the whole paper for this problem.

6.      You have not acknowledged the source of some equations. E.g. equations 1 and 2. Make sure you acknowledge the source of the equations by citing their original sources.  

Moderate editing of English language

Reviewer 2 Report

Ms. Sustainability MDPI

Title: A comprehensive review of Electric vehicle charging stations 2 with solar photovoltaic system

 General recommendation:

In this manuscript review, the authors have review of Electric vehicle charging stations 2 with solar photovoltaic system. I found the paper to be overall well written and much of it to be well described. Therefore, I am sure that the current review is on a topic of relevance and general interest to the readers of the journal. However, I still recommend a minor revision is warranted. I explain my concerns in detail below

1.       The position of all Figures in the manuscript should be aligned to be attractive to the reader

2.       The authors should maintain consistency in the formatting of Figures 1-8 and also Table 1- 6 in the manuscript, some are slanted, and others are upright. It should be written uniformly

3.       There are two Table 2 in the manuscript, without Table 4. The authors need to be corrected.

4.       The information and discussion presented are limited. The are also very few conclusions in this review paper. The author needs to add more comprehensive information from this review, both in the discussion and in the conclusion.

Reviewer 3 Report

Do not use acronyms in chapter titles and subtitles. For example, at chapter 2.3.2. Wireless Charging (WC), the acronym WC is very weird, sounds like toilet.

In table 1 delete the units of measure written after the power values. The unit of measure is entered at the head of the table (Power kW). Also, in table 1, at type of vehicle it is not clear what 4w, 3w, 2w means.

Combine Chapter 5 with Chapter 6 into one, Discussion and Conclusions.

Reviewer 4 Report

In this article, the author introduces using the current utility grid, which is powered by the fossil fuel basing generating system, to charge EVs has an impact on the distribution system and could not be ecologically beneficial. The current electric vehicle (EV) market, technical requirements, charging infrastructure, and grid implications are all examined in depth in this paper. The report gives overview of present EV situation as well as a thorough analysis of significant global EV charging and grid connectivity standards. Finally, the challenges and suggestions for future expansion of the infrastructure of EV charging, grid integration, are evaluated and summarised. However, this manuscript has some shortcomings and should be further revised before it is considered for publication. Let us elaborate on some of them:

1. The format of the article, especially the recently 3 years reference.

2. I think the graphic abstract should be added.

3. The references cited in the article should be relevant to the content of the article. Please quote reasonably. At the same time, the author is requested to standardize the format of references. Cited the model and artificial intelligence methods in energy element such as DOI: 10.1016/j.est.2021.103099 10.3390/en16073167; 10.3390/en16041599.

4. The figure and table caption should be more informative.

5.What’s the innvotion of this article? The author should explain.

6. I suggest the author tell the novelty of the article and suggest to compare with others.

In short, in its current form, the paper is not suitable for acceptance. The paper needs further modification, by addressing the above-mentioned comments.

 Minor editing of English language required

Round 2

Reviewer 1 Report

The authors respond to the comments appropriately 

Minor editing of English language required